# Phylogeny and Systematics of the Genus *Tolypocladium* (Ophiocordycipitaceae, Hypocreales)

**DOI:** 10.3390/jof8111158

**Published:** 2022-11-01

**Authors:** Quan-Ying Dong, Yao Wang, Zhi-Qin Wang, Yan-Fang Liu, Hong Yu

**Affiliations:** 1Yunnan Herbal Laboratory, College of Ecology and Environmental Sciences, Yunnan University, Kunming 650504, China; 2The International Joint Research Center for Sustainable Utilization of Cordyceps Bioresources in China and Southeast Asia, Yunnan University, Kunming 650504, China

**Keywords:** micromorphology, phylogenetic analyses, taxonomy, three new taxa

## Abstract

The taxonomy and phylogeny of the genus *Tolypocladium* are herein revised based on the most comprehensive dataset to date. Two species-level phylogenies of *Tolypocladium* were constructed: a single-gene phylogeny (ITS) of 35 accepted species and a multigene phylogeny (nr*SSU*, nr*LSU*, *tef*-*1α*, *rpb1,* and *rpb2*) of 27 accepted species. Three new species, *Tolypocladium pseudoalbum* sp. nov., *Tolypocladium subparadoxum* sp. nov., and *Tolypocladium yunnanense* sp. nov., are described in the present study. The genetic divergences of four markers (ITS, *tef*-*1α*, *rpb1* and *rpb2*) among *Tolypocladium* species are also reported. The results indicated that species of *Tolypocladium* were best delimited by *rpb1* sequence data, followed by the sequence data for the *rpb2*, *tef*-*1α*, and ITS provided regions. Finally, a key to the 48 accepted species of *Tolypocladium* worldwide is provided.

## 1. Introduction

*Tolypocladium* was originally described as an anamorph genus by Gams in 1971 to accommodate three species collected from soil: *T. cylindrosporum* W. Gams, *T. geodes* W. Gams, and *T. inflatum* W. Gams [1]. Subsequently, the species *T. lignicola* G.L. Barron*, T. parasiticum* G.L. Barron, and *T. trigonosporum* G.L. Barron, all of which were isolated from bdelloid rotifers, were added to this genus [2,3,4]. Bissett described *T. nubicola* and *T. tundrense* from soil in 1983 [5] and reassigned three species to *Tolypocladium*: *T. balanoide* (basionym: *Cephalosporium balanoide*)*, T. microsporum* (basionym: *Verticillium microsporum*) and *T. niveum* (basionym: *Pachybasium niveum*). Additionally, Bissett [5] noted that the morphological characteristics of *T. niveum* were similar to those of *T. inflatum.* Because *T. niveum* precedes *T. inflatum*, Bissett proposed that *T. inflatum* be synonymized with *T. niveum* [5]. However, Dreyfuss observed that *T. inflatum* produces cyclosporine and is the type species of the genus *Tolypocladium*. The name *T. inflatum* is also commonly accepted [6]. Therefore, Dreyfuss rejected the synonymization of *T. inflatum* with *T. niveum* [6]. The genus *Tolypocladium* is morphologically characterized by sparingly branched conidiophores, swollen phialides, and one-celled conidia borne in slimy heads. Approximately 20 species have been included in the *Tolypocladium* based on morphological characteristics.

The taxonomy of *Tolypocladium* has been discussed extensively for decades. *Cordyceps* sensu lato was recently reclassified into three families (Clavicipitaceae *sensu stricto*, Cordycipitaceae, and Ophiocordycipitaceae) and four genera (*Cordyceps* s. str., *Elaphocordyceps*, *Metacordyceps*, and *Ophiocordyceps*) based on multigene phylogeny [7]. Molecular phylogenetic analyses suggested that *Tolypocladium* species fall within the Ophicordycipitaceae [7,8]. The genus *Elaphocordyceps* Sung and Spatafora 2007 was proposed for 23 species of the *Cordyceps* Fr. (1818: 316); these species parasitize the fungal genus *Elaphomyces* and some species of arthropods (e.g., cicada nymphs and beetle larvae) [7]. The *Elaphocordyceps* species within the Ophiocordycipitaceae form a clade sister to those of the genus *Ophiocordyceps*. Gams established the *Chaunopycnis* to accommodate *C. alba*, which morphologically resembles *Tolypocladium* species in its conidiogenesis [9]. With the end of dual nomenclature for fungi, the generic name *Tolypocladium* was chosen over *Elaphocordyceps* and *Chaunopycnis* as *Tolypocladium* is the oldest and most commonly used name [8]. *Chaunopycnis* was integrated into the genus *Tolypocladium*. Accordingly, *C. alba*, *C. ovalispora*, and *C. pustulata* were renamed *T. album*, *T. ovalisporum*, and *T. pustulatum*, respectively [8].

At present, 53 *Tolypocladium* records, including 5 varieties, are listed in the *Index Fungorum* (www.indexfungorum.org, accessed on 28 August 2022). *Tolypocladium balanoides,* which was reassigned to *Drechmeria* (as *Drechmeria balanoides*), and *Tolypocladium parasiticum,* which was reassigned to *Metapochonia* (as *Tolypocladium parasiticum*), should be excluded from the *Tolypocladium*. However, some of these records are doubtful, because the original identifications were presumptive based on host associations or based on the morphology of only one or two ascospore stages of the asexual or sexual morph. For 16 species, no molecular data are available in the GenBank database [10]. *Tolypocladium* species have a cosmopolitan distribution and a broad host range that includes bdelloid rotifers, mosquito larvae, nematodes, fireflies, beetles, cicada nymphs, batmoth larvae, macrocystic fungi, *Ophiocordyceps sinensis,* and even plants (as endophytes) [2,3,11,12,13,14,15,16,17,18,19].

*Tolypocladium* species have been widely studied due to their importance in the medicinal domain. These species can produce cyclosporine A, tolypoalbin, tolypin, cyclosporine D hydroperoxide, cylindromicin, and tolyprolinol [20,21], all of which have significant antitumoral, anti-inflammatory, antifungal, and/or antiparasitic properties [22]. Cyclosporine A, which is naturally isolated from *T. inflatum*, is widely used in autoimmune disease treatment and to prevent allograft rejection [23,24,25]. Tolypoalbin is a peptide mixture and a tetrameric acid produced by *T. album* [26]. Tolypin is also a peptide mixture [27]. Like kojic acid, cylindromicin is a significant bioactive inhibitor of tyrosinase [28]. Tolyprolinol, a dipeptide produced by *Tolypocladium* sp. FKI-7981, contains a rare moiety prolinol and was the first natural product isolated from *Tolypocladium* species. Tolyprolinol exhibits moderate antimalarial activity without cytotoxicity or any other antimicrobial properties [29].

Recent investigations and phylogenetic analyses have ascribed many new taxa to *Tolypocladium*. Therefore, the diversity of *Tolypocladium* may be underestimated. In the present study, we aimed first to investigate and document the worldwide diversity of *Tolypocladium* fungi using our current collection of specimens and data collected over the last several years. We used comprehensive morphological and molecular phylogenetic reconstructions to identify and reevaluate our specimens. Based on these reconstructions, we herein describe and illustrate three new taxa. We then clarify the phylogenetic affinities of these new taxa using rDNA sequence analyses.

## 2. Materials and Methods

### 2.1. Sampling

*Tolypocladium* species were collected in Kunming, Pu’er, Yunnan, China. Voucher specimens and the corresponding isolated strains were deposited in the Yunnan Herbal Herbarium (YHH) and the Yunnan Fungal Culture Collection (YFCC), respectively, of Yunnan University, Kunming, China.

*Tolypocladium* strains were isolated from soil samples, as described in our previous publication [30]. In brief, 2 g of soil was added to a flask containing 20 mL of sterilized water and glass beads. The suspension was then shaken for 10 min and diluted 100 times. Finally, 200 µL of diluted soil suspension was spread on petri dishes containing solidified onion garlic agar (OGA: 1 L of distilled water, 20 g of grated garlic, and 20 g of onion were boiled together for 1 h; the boiled biomass was filtered and 2% agar was added to the filtrate). Czapek yeast extract agar (CYA; Advanced Technology and Industrial Co., Ltd., Hong Kong, China) and potato dextrose agar (PDA; Difco, USA) were used. Rose bengal (50 mg/L) and kanamycin (100 mg/L) were added to all media. Conidia grown on insect cadavers were transferred to PDA plates and cultured at 22 °C. The filamentous fungal colonies isolated from the culture were transferred to fresh PDA media. The purified fungal strains were maintained at 22 °C in a culture room or transferred to PDA slants and stored at 4 °C.

### 2.2. Morphological Studies

Morphological studies were performed as described in our previous study [31]. Micromorphological characteristics, such as phialides and conidia, were studied by picking and mounting cultures on glass slides. The sizes and shapes of the microcharacteristics were determined using an Olympus CX40 and BX53 (Olympus Corporation, Tokyo, Japan). Individual length and width measurements were taken for 20–30 replicates, including the absolute minima and maxima. The morphological characteristics were described based on the digital images and the measurement dataset.

### 2.3. Molecular Studies

#### 2.3.1. DNA Extraction and PCR Amplification

Total DNA was extracted from the fungal mycelia on PDA plates or from herbarium materials using the modified CTAB procedure [32]. The primer pair nr*SSU*-CoF and nr*SSU*-CoR [33] was used to amplify nr*SSU*, the primer pair LR5 and LR0R [34,35] was used to amplify nr*LSU*, and the primer pair EF1α-EF and EF1α-ER [7,36] was used to amplify the translation elongation factor 1α (*tef-1α*). The primer pair RPB1-5′F and RPB1-5′R and the primer pair RPB2-5′F and RPB2-5′R [7,36] were used to amplify the largest and second-largest subunits of RNA polymerase II (*rpb1* and *rpb2*), respectively. The ITS fragment was amplified using the primer pair ITS5 and ITS4 [37].

The matrix for the polymerase chain reaction (PCR) was comprised of 2.5 μL PCR 10× buffer (2 mmol/L Mg^2+^) (Transgen Biotech, Beijing, China), 1 μL forward primer (10 µmol/L), 1 μL reverse primer (10 µmol/L), 0.25 μL Taq DNA polymerase (Transgen Biotech, Beijing, China), 2 μL dNTP (2.5 mmol/L), 1 μL DNA template (500 ng/μL), and 17.25 μL sterile ddH_2_O. Amplification reactions were performed in a Bio-Rad T100 thermal cycler (Bio-Rad Laboratories, CA, USA). The PCR cycling conditions for the amplification of nr*SSU* were as follows: 95 °C for 4 min; eight cycles of 94 °C for 50 s, 56 °C for 50 s, and 72 °C for 2 min, with the annealing temperature decreasing 0.5 °C/cycle; 25 cycles of 94 °C for 50 s, 52 °C for 50 s, and 72 °C for 2 min; and 72 °C for 10 min. The nucleotide sequences of ITS, nr*LSU*, *tef-1α*, *rpb1*, and *rpb2* were amplified using the following cycling conditions: 95 °C for 4 min; eight cycles of 94 °C for 50 s, 56 °C for 50 s, and 72 °C for 70 s, with the annealing temperature decreasing 0.5 °C/cycle; 25 cycles of 94 °C for 50 s, 52 °C for 50 s, and 72 °C for 70 s; and 72 °C for 10 min. PCR products were purified using a gel extraction and PCR purification combo kit (Beijing Genomics Institute, Shenzhen, China) and sequenced on an automatic sequence analyzer (BGI Co., Ltd., Shenzhen, China) using the amplification primers.

#### 2.3.2. DNA Sequence Alignments

To investigate the placement of our samples within *Tolypocladium,* the nucleotide sequences of ITS, nr*SSU*, nr*LSU*, *tef*-*1α*, *rpb1,* and *rpb2* were compared with sequences from representative *Tolypocladium* species downloaded from GenBank (Table 1, Figure 1 and Figure 2). Individual gene sequence datasets (ITS, nr*SSU*, nr*LSU*, *tef*-*1α*, *rpb1,* and *rpb2*) were aligned and manually checked using Bioedit v7.0.9 [38]. To identify possible phylogenetic conflicts among the datasets, the partition homogeneity (PH) test was performed with 1000 randomized replicates of heuristic searches with simple sequence addition in PAUP* 4.0a166 (http://paup.phylosolutions.com, accessed on 28 August 2022) [39]. The results showed that the phylogenetic signals from the five gene markers were in conflict.

#### 2.3.3. Phylogenetic Analyses

Phylogenetic analyses were based on a concatenated five-gene dataset and the ITS sequences alone. nr*SSU*, nr*LSU*, *tef*-*1α*, *rpb1, rpb2,* and ITS sequences were retrieved from GenBank, and combined with those generated in this study. Taxon information and GenBank accession numbers are given in Table 1. Sequences were aligned using Clustal X2.0 and MEGA v6.06 [40,41]. Group I introns in the nr*SSU* sequences of some species were excluded from the phylogenetic analyses, and gaps were treated as missing data. After alignment of the five genes individually, the alignments were concatenated. A partition homogeneity test was conducted in PAUP* 4.0a166 [39], and the results indicated that there were no conflicts among the data partitions. PartitionFinder V1.1.1 identified eleven data partitions: nine corresponding to the three codon positions in each of the protein-coding genes (*tef*-*1α*, *rpb1,* and *rpb2*) and one each for nr*LSU* and nr*SSU* [42,43]. The results showed that the phylogenetic signals of the five genes were congruent (*p* = 0.02).

Maximum likelihood (ML) phylogenetic analyses were conducted using RaxML 7.0.3 [44] with the recommended partition parameters and 1000 rapid bootstrap replicates. Bayesian posterior probabilities (BP) were estimated with the same partition parameters using MrBayes v3.1.2 [45]. Bayesian inference (BI) analysis ran in MrBayes v3.1.2 for 5 million generations. Maximum parsimony (MP) analysis of the ITS dataset was performed using PAUP v. 4.0a166 [39], adopting the random addition of sequences model (10 replications), with gaps treated as missing data. A bootstrap (MPBS) analysis was performed using the maximum parsimony criterion in 1000 replications.

The following taxa were included in the five-gene concatenated dataset: *Drechmeria* W. Gams and H.-B. Jansson, *Harposporium* Lohde, *Ophiocordyceps* Petch, *Purpureocillium* Luangsa-Ard, Hywel-Jones, Houbraken and Samson, and *Tolypocladium*. Two species of *Polycephalomyces* Kobayasi were used as outgroups. ITS analysis was performed on *Tolypocladium* taxa only. Phylogenetic trees were visualized with FigTree v1.4.0 [46], edited in Microsoft PowerPoint, saved in PDF format, and converted to JPG format using Adobe Illustrator CS6 (Adobe Systems Inc., San Jose, USA). The finalized alignments and trees were submitted to TreeBASE (multigene submission ID 29808).

We calculated a phylogenetic distance matrix for the markers ITS, *tef*-*1α*, *rpb1,* and *rpb2* to assess the species boundaries of the 10 *Tolypocladium* species (Appendix A), because the sequence data were complete for these four loci. The paired distances among the 10 *Tolypocladium* lineages were measured using the Kimura two-parameter model in MEGA v6.06 [41].

## 3. Results

### 3.1. Sequence Alignment and Phylogenetic Analyses

ITS, nr*SSU*, nr*LSU*, *tef-1α*, *rpb1*, and *rpb2* sequences were generated from ten living cultures (accession numbers are given in Table 1). The concatenated five-gene alignment of 113 taxa contained 5371 base pairs in total: nr*SSU*, 1488 bp; nr*LSU*, 987 bp; *tef-1α,* 998 bp; *rpb1,* 756 bp; and *rpb2,* 1142 bp. *Polycephalomyces formosus* ARSEF 1424 and *Polycephalomyces sinensis* CN 80-2 were used as the outgroup sequences for the five-gene phylogenetic analyses. Both BI and ML analyses recovered six well-supported clades corresponding to the *Ophiocordyceps* (ML bootstrap, BS = 85% and bayesian posterior probability, BP = 1), *Tolypocladium* (BS = 99%, BP = 1), *Purpureocillium* (BS = 97%, BP = 1), *Drechmeria* (BS = 97%, BP = 1), *Harposporium* (BS = 88%, BP = 1)*,* and *Polycephalomyces* (BS = 100%, BP = 1) (Figure 1) within Ophiocordycipitaceae. Phylogenetically, the *Tolypocladium* clade is the closest to the *Ophiocordyceps* clade, and it is well supported in this and other published analyses [7,8]. According to the current data, relationships for species in the *Tolypocladium* clade show strong statistical support for internal branches. Most sexual species are located at the top of the *Tolypocladium* clade, and asexual species are located at the bottom of the *Tolypocladium* clade, except *T. subparadoxum* and *T. paradoxum*. Three new species (i.e., *Tolypocladium pseudoalbum* sp. nov., *Tolypocladium subparadoxum* sp. nov., and *Tolypocladium yunnanense* sp. nov.) were recognized in *Tolypocladium* (shown in boldface in Figure 1). *T. pseudoalbum* sp. nov. formed a clade with *T. pustulatum*, *T. tropicale*, *T. endophyticum*, *T. amazonense*, and *T. yunnanense* sp. nov. (Figure 1), while *T. subparadoxum* sp. nov. formed a well-supported clade with *Tolypocladium* sp. and *T. paradoxum* (Figure 1). *T. yunnanense* sp. nov. was close to five other species: *T. pustulatum*, *T. tropicale*, *T. endophyticum*, *T. amazonense*, and *T. pseudoalbum* sp. nov. (Figure 1).

The ITS dataset used for phylogenetic analyses comprised 769 base pairs of sequence data for 61 taxa. *Purpureocillium lilacinum* CBS 284.36 and *Purpureocillium lilacinum* NHJ 3497 were chosen as outgroup sequences. The three phylogenetic algorithms (BI, ML, and MP) recovered trees with similar topologies (Figure 2). The three new species described herein (i.e., *Tolypocladium pseudoalbum* sp. nov., *Tolypocladium subparadoxum* sp. nov., and *Tolypocladium yunnanense* sp. nov.) formed an independent lineage with *Tolypocladium* (Figure 2).

### 3.2. Genetic Distance Analyses

Comparisons of genetic divergence showed that (1) the minimum thresholds (p-distances) required to distinguish species within the *Tolypocladium* lineages were 0.026, 0.017, 0.013, and 0.008 for *tef*-*1α*, *rpb1*, *rpb2*, and ITS, respectively (Appendix A); and (2) the phylogenetic relationships within *Tolypocladium* were best resolved by the *rpb1* sequence data, followed by those of *rpb2, tef*-*1α,* and ITS (Appendix A).

### 3.3. Taxonomy

***Tolypocladium*** W. Gams, Persoonia 6(2): 185 (1971). emend. C. A. Quandt et al. IMA Fungus 5: 125 (2014).

**Synonyms**: *Chaunopycnis* W. Gams, Persoonia 11: 75 (1980).

*Elaphocordyceps* G. H. Sung and Spatafora, Stud. Mycol. 57: 36 (2007).

**Sexual morph:** Stromata are solitary or several, simple or branched. The stipe is tough, dark-brownish to greenish, cylindrical, and abruptly to enlarging in the fertile part. The fertile part is cylindrical to clavate. Perithecia are superficial, wholly or partially immersed, ordinal or oblique in arrangement. Asci are cylindrical with a thickened ascus apex. Ascospores are usually cylindrical, multiseptate, disarticulate into part spores, and are occasionally non-disarticulating. Part spores are cylindrical.

**Asexual morph:** *Tolypocladium*-like, *Chaunopycnis*-like, or *Verticillium*-like. Conidiophores typically are short and bear whorls of phialides. Phialides often have bent necks and are usually swollen at the base. Conidia are ellipsoidal, globose, or reniform, and aggregate in small heads at the tips of the phialides.

***Tolypocladium pseudoalbum***, H. Yu, Y. Wang and Q.Y. Dong, sp. nov., Figure 3.

**MycoBank:** MB 845430.

**Etymology:** Referring to the morphological resemblance of this species to *Tolypocladium album,* despite its phylogenetic dissimilarity.

**Type:** China, Yunnan Province, Kunming City, Wild Duck Forest Park (25°13′ N, 102°87′ E, 2100 m above sea level), from the soil on the forest floor, 10 August 2019, Yao Wang (holotype: YHH 875, dried specimen; ex-type living culture: YFCC 875).

**Teleomorph:** Unknown.

**Anamorph:** Colonies on PDA are moderately fast-growing, attaining a diameter of 42–44 mm in 21 days at 22 °C. Colonies pulvinate, with high mycelial density, white or pale yellow, reverse deep yellow. Hyphae branched, smooth-walled, septate, hyaline, 1.1–2.7 μm wide. Cultures readily produce phialides and conidia on PDA after two weeks at room temperature. Phialides arising from aerial hyphae, solitary, 12.3–48.5 × 1.0–2.0 μm, cylindrical, tapering gradually toward the apex, neck 1.4–4.6 × 0.8–1.8 µm. Conidia hyaline, one-celled, globose to broadly ellipsoidal 1.8–3.4 × 1.3–1.9 μm. Chlamydospores present.

**Habitat:** Soil.

**Known distribution:** China.

**Additional specimens examined:** China, Yunnan Province, Kunming City, Songming County, Dashao Village (25°23′ N, 102°33′ E, 2700 m above sea level), from the soil on the forest floor, 12 August 2018, Yao Wang (living culture: YFCC 876).

**Comments:** Five species are closely related to *T. pseudoalbum* sp. nov., i.e., *T. pustulatum*, *T. tropicale*, *T. endophyticum*, *T. amazonense*, and *T. yunnanense* sp. nov. This clade is characterized by cylindrical to lageniform phialides, globose to broadly ellipsoidal conidia, and primarily white colonies. The phialides of *T. pseudoalbum* sp. nov. (12.3–48.5 × 1.0–2.0 μm) are longer than those of *T. album* (3.5–10 × 1.0–1.5 µm).

***Tolypocladium subparadoxum*** H. Yu, Y. Wang and Q.Y. Dong, sp. nov., Figure 4.

**MycoBank:** MB 845431.

**Etymology:** Referring to the phylogenetic placement is closely related to *T. paradoxum*.

**Holotype:** China, Yunnan Province, Pu’er City, Simao District (22°43′ N, 100°58′ E, 1360 m above sea level), from soil on the forest floor, 27 August 2021, Yao Wang (holotype: YHH 879, dried specimen; ex-type living culture: YFCC 879).

**Teleomorph:** Not observed.

**Anamorph:** Colonies on PDA are moderately fast-growing, attaining a diameter of 36–38 mm in 21 days at 22 °C. Colonies flocculent, fluffy, with low mycelial density, white or pale yellow, reverse deep yellow. Hyphae smooth-walled, branched, septate, hyaline, 0.8–2.2 μm wide. Cultures produce phialides and conidia on PDA after two weeks at room temperature. Phialides arising from aerial hyphae, solitary, or in verticils of two to four, 5.4–40.1 × 0.9–1.8 μm, cylindrical, tapering gradually toward the apex, neck 3.2–5 × 0.7–1.2 µm. Conidia hyaline, one-celled, ellipsoidal or globose, single or aggregating in heads at the apex of phialides, 2.6–6.5 × 1.0–2.9 μm. Chlamydospores not observed.

**Habitat:** Soil, larvae of cicada.

**Known distribution:** China, Japan.

**Additional specimens examined:** NBRC 106958, Niryo, Takatsuki-shi, Osaka Prefecture.

**Comments:** Our phylogenetic analysis indicates that *Tolypocladium subparadoxum* sp. nov. is closely related to *Tolypocladium* sp. and *T. paradoxum*. The two strains (YFCC 879 and NBRC 106958) formed a distinct lineage. NBRC 106958 was firstly isolated from cicada in Japan by S. Ban (https://www.nite.go.jp/nbrc/catalogue/NBRCCatalogueDetailServlet?ID=NBRCandCAT=00106958, accessed on 28 August 2022) and subsequently isolated from soil in China (YFCC 879). Since no significant morphological differences were found between the Chinese collections and that of Japan (Appendix A), we treated YFCC 879 and NBRC 106958 as *Tolypocladium subparadoxum*. *Tolypocladium paradoxum* was originally described as *Cordyceps paradoxa* by Kobayasi, which was a cicada pathogen that produces solitary, pale ochraceous to dark olivaceous, fleshy stromata with cylindrical asci, breaking into cylindrical part spores [76]. Morphologically, *T. subparadoxum* differs from *T. paradoxum* in the following aspects. Relatively*, T. paradoxum* has longer phialides measured 5.8–58.3 × 1.8–4.3 µm, broader neck (0.9–1.9 µm vs 0.7–1.2 µm), and minor conidia (2.3–4.8 × 1.9–5.2 µm vs 2.6–6.5 × 1.0–2.9 μm) (Appendix A).

*Tolypocladium subparadoxum* similar to *T. dujiaolongae* and sharing cicada host, solitary, or verticillate, cylindrical or conical phialides, globose to ovoid conidia, and conidia aggregating mostly in small heads, but the latter differs by its relatively shorter phialides (11–35 × 1.0–2.7 μm vs 5.4–40.1 × 0.9–1.8 μm) [19]. Our phylogenetic analysis inferred from ITS data (Figure 2) suggests that they represent two distinct species.

*Tolypocladium geodes* is also similar to *T. subparadoxum* in their soil habitats and ellipsoidal or globose conidia. However, *T. geodes* has relatively shorter phialides (5.6–12.4 × 1.4–2.4 µm) and somewhat minor conidia (1.9–2.4 × 1.6–2.0 µm) [5]. Molecular phylogenetic analyses (Figure 1 and Figure 2) indicate that they are distinct species.

***Tolypocladium yunnanense*** H. Yu, Y. Wang and Q.Y. Dong, sp. nov., Figure 5

**MycoBank:** MB 845432.

**Etymology:***Yunnanense* (Lat.) refers to the type locality (Yunnan, China).

**Holotype:** China, Yunnan Province, Kunming City, Wild Duck Forest Park (25°14′ N, 102°87′ E, 2080 m above sea level), from soil on the forest floor, 12 August 2018, Yao Wang (holotype: YHH 877, dried specimen; ex-type living culture: YFCC 877).

**Teleomorph:** Unknown.

**Anamorph:** Colonies on PDA are moderately fast-growing, attaining a diameter of 44–46 mm in 21 days at 22 °C. Colonies pulvinate, with high mycelial density, whitish to orange-yellow, reverse deep yellow. Hyphae smooth-walled, branched, septate, hyaline, 1.0–2.4 μm wide. Cultures produce phialides and conidia on PDA after two weeks at room temperature. Phialides are usually curved, solitary, 7.6–62.6 × 0.9–2.3 μm, cylindrical, narrowing slightly or abruptly into a neck, 3–4.2 × 0.5–1 µm. Conidia hyaline, one-celled, elliptical to subglobose, 1.2–2.4 × 0.9–1.9 μm. Chlamydospores present.

**Habitat:** Soil.

**Known distribution:** China.

**Additional specimens examined:** China, Yunnan Province, Pu’er City, Simao District (22°42′ N, 100°57′ E, 1348 m above sea level), from soil on the forest floor, 7 October 2019, Yao Wang (living culture: YFCC 878).

**Comments:***Tolypocladium yunnanense* sp. nov. is characterized by its solitary cylindrical phialides (7.6–62.6 × 0.9–2.3 μm), elliptical to subglobose conidia (1.2–2.4 × 0.9–1.9 μm), and white colonies. The five-gene phylogenetic analysis suggested that *T. yunnanense* sp. nov. was closely related to five other species (*T. pustulatum*, *T. tropicale*, *T. endophyticum*, *T. amazonense* and *T. pseudoalbum* sp. nov.). Phylogenetic analyses of this clade using ITS sequences, for which more complete data were available, showed that *T. yunnanense* sp. nov. formed clade with *T. album*, *T. pseudoalbum* sp. nov., *T. tropicale*, *T. amazonense,* and *T. endophyticum*. Morphologically, *Tolypocladium yunnanense* sp. nov. has longer phialides than other species in this clade: *Tolypocladium yunnanense* sp. nov., 7.6–62.6 × 0.9–2.3 μm; *T. pustulatum*, 4–10 × 2–4 µm, *T. tropicale*, 4.6 × 1.5 µm; *T. endophyticum*, 4.1 × 1.6 µm; *T. amazonense*, 4.1 × 1.6 µm; *T. pseudoalbum* sp. nov., 12.3–48.5 × 1.0–2.0 μm, and *T. album,* 3.5–10 × 1.0–1.5 µm.


**Key to *Tolypocladium* species worldwide**
Sexual state observed…………………………………………………………………………………………………………………………..1Sexual state not observed…………………………………………………………………………………………………………………..27 1a. Perithecia superficial or half-immersed ……………………………………………………………………………………………..2 1b. Perithecia completely immersed………………………………………………………………………………………3 2a. Perithecia pyriform, relatively larger, 520–550 × 260–280 µm, asci relatively larger, 400–450 × 7–7.5 µm, part spores 2.5–3.0 × 3.0 µm, on cicada nymphs, stromata relatively longer, 14 cm long……………………………………………………………………………………………………………***T. inegoense*** 2b. Perithecia ovoid, relatively smaller, 320–380 × 220–280 µm, asci cylindrical, smaller, 240–250 × 6 µm, not dissociate into part spores, on *Elaphomyces*, stromata shorter, 3.5–4.5 cm long……………………………***T. ramosum*** 3a. Perithecia ellipsoid, subglobose to ovoid…………………………………………………………………………………….4 3b. Perithecia ampullaceous……………………………………………………………………………………………………………………………..26 4a. From multiple substrate/host (beetle or moth larvae, Larvae of *Scarabaeidae* (sexual morph); soil, humus, *Picea glauca*, roots of *Picea mariana*, the surface of *Mycobates* sp. (*Acari*, *Mycobatidae*), the sclerotium of *Ophiocordyceps gracilis* (asexual morph)………………………………………………………………….………..***T. inflatum*** 4b. From simple substrate/host…………………………………………………………………………………………….5 5a. On beetle or unidentified host……………………………………………………………………………………………6 5b. On *Elaphomyces*…………………………………………………………………………………………………………..9 6a. On the unidentified host, asci relatively wider, 10–15 µm…………………………………………………***T. cucullae*** 6b. On beetle, asci narrower than 10 µm………………………………………………………………………………….7 7a. Stromata was connected to the host through a yellowish rhizomorph-like structure………………..***T. fumosum*** 7b. Stromata arising directly from the host, never rhizomorphic……………………………………………………….8 8a. Part spores short cylindrical, truncate at both ends, 3–5 × 1.5–2 µm……………………………….…***T. paradoxum*** 8b. Part spores very short, almost cuboid in side view, without flattened ends, 1.5–2.5 × 1.5–1.7 µm…………………………………………………………………………………………………***T. toriharamontanum*** 9a. Stromata clavate, the fertile part not abruptly enlarged from the stipe………………………………………….10 9b. Stromata capitate, the fertile part spherical, oval or cylindrical abruptly enlarged from the stipe………………17 10a. Stromata size relatively larger, 10–12 cm long………………………..…………………………………***T. jezoense*** 10b. Stromata size < 10 cm………………………………………………………………………………………………….11 11a. Part spores articulate, moniliform……………………………………………………………………***T. szemaoense*** 11b. Part spores cylindrical……………………………………………………………………………………………….12 12a. Stromata was connected to the host through a rhizomorph-like structure……………………………………………..13 12b. Stromata arising directly from the host, never rhizomorphic…………………………………………………….14 13a. Fertile part yellowish-green when young, turning olive-green as it matures, perithecia relatively smaller, 480–590 × 195–235 μm…………………………………………………………………………………***T. bacillisporum*** 13b. Fertile part reddish brown to olivaceous brown, perithecia larger, 600–800 × 250–500 µm…***T. ophioglossoides*** 14a. Fertile part black, yellow black, dark chestnut brown when dried……………………………..………………15 14b. Fertile part pale bluish to grayish blue………………………………………………………***T. valvatistipitatum*** 15a. Perithecia ≤ 700 µm long…………………………………………..…………………………………………………16 15b. Perithecia > 700 µm long (750–1000 × 250–300 µm)……………..…………………………………***T. tenuisporum*** 16a. Perithecia relatively narrower, 567–697 × 206–248 µm, part spores smaller, 2–5 × 1.5–2 µm, stromata 1.5–3 cm long…………………………………………………………………………………………………..……***T. flavonigrum*** 16b. Perithecia relatively wider, 500–700 × 250–350 µm, part spores larger, 10–18 × 2.5–4 µm, stromata 2.5–7 cm long………………………………………………………………………………………………………..…***T. japonicum*** 17a. Perithecia larger………………………………………………………………………………………………………18 17b. Perithecia smaller, 400 × 250 µm…………………………………………………………………….………***T. virens*** 18a. Stromata 12 cm long, part spores very long, 40–65 µm long.………………………………..***T. longisegmentatum*** 18b. Stromata shorter than 12 cm, part spores < 40 µm long……………………………………………………………19 19a. Part spores ≤ 8 µm long……………………………………………………………………………………………….20 19b. Part spores > 8 µm long…………………………………………………………………………………………….…22 20a. Asci shorter than 300 µm (240–300 × 7–8 µm), perithecia relatively smaller (450–540 × 230–260 µm).…………………………………………………………………………………………………….…***T. intermedium*** 20b. Asci longer than 300 µm, perithecia larger……………………………………………………………………..…21 21a. Stipe slender, 0.5–1.0 mm thick, yellowish green to olivaceous, stromata shorter, 1.5–2.5 cm long, part spores, 2–5 × 1.5–2 µm…………………………………………………………………………………………………***T. fractum*** 21b. Stipe 1–5 mm thick, dark brown, smooth or furfuraceous, stromata 5–7 cm long, part spores longer, 3–8 × 2 µm……………………………………………………………………………………………………………***T. valliforme*** 22a. Perithecia < 550 µm long (480–540 µm)……………………………………………………….***T. delicatistipitatum*** 22b. Perithecia > 550 µm long……………………………………………………………………………………………..23 23a. Part spores < 15 µm long (8–11 µm)………………………………………………………………..***T. miomoteanum*** 23b. Part spores ≥ 15 µm long…………………………………………………………………………….……………….24 24a. Part spores < 3 µm wide……………………………………………………………………………………………..25 24b. Part spores ≥ 3 µm wide (3.0–4.5 µm)…………………………………………………………***T. inusitaticapitatum*** 25a. Fertile part olive-brown to olive-black, perithecia relatively larger, 650–950 × 250–420 µm, asci wider, 350–540 × 10–12 μm, part spores cylindrical or somewhat fusoid, 8–25 × 2.5–3 µm……………..……………***T. capitatum*** 25b. Fertile part purple-brown, blacker when older, perithecia smaller, 600–750 × 200–300 µm, asci slender, 350–500 × 8–10 µm, part spores filiform, spindle-shaped, 15–20 × 2–3 µm……………………………….***T. rouxii*** 26a. Perithecia relatively shorter, 520–740 × 300–330 μm, part spores cylindrical, 3–7 × 2–3 μm, on cicada nymphs……………………………………………………………………………………………………***T. dujiaolongae*** 26b. Perithecia relatively longer, 900–930 × 220–250 µm, part spores fusoid, 16–18 × 3 µm, on *Elaphomyces*…………………………………………………………………………………………………………***T. minazukiense*** 27a. From multiple substrate/host…………………………………………………………………………………..……28(*T. album, T. cylindrosporum*, *T. inflatum*, *T. pustulatum*, *T. subparadoxum*) 27b. From only a type of substrate/host………………………………………………………………………………….31 28a. Phialides cylindrical…….……………………………………………………………………………………………29 28b. Phialides ellipsoidal to subglobos…………………………………………………………….…***T. cylindrosporum*** 29a. Colonies white, conidia globose to ovoid (phialides 3.5–10 × 1–1.5 µm, conidia 3.5 × 1.5–2.0 µm)……………………………………………………………………………………………………….…………***T. album*** 29b. Colonies white to pale yellow, conidia ellipsoidal, globose or broadly ellipsoidal……………………………30 30a. Phialides 4–10 × 2–4 µm, conidia 2–3 × 1.5–2.5 µm…………………………………………………***T. pustulatum*** 30b. Phialides 5.4–40.1 × 0.9–1.8 μm, conidia larger, 2.6–6.5 × 1–2.9 μm……………………………***T. subparadoxum*** 31a. From substrate…………………………………………………………………………………………………….….32 31b. On insects……………………………………………………………………………………………………………..45 32a. Substrate is not fungus………………………………………………………………………………………………33 32b. Substrate is fungus……………………………………………………………………………………………………43 33a. From plant tissue……………………………………………………………………………………..………………34 (*T. amazonense*, *T. endophyticum*, *T. ovalisporum*, *T. tropicale*) 33b. From soil…………………………………………………………………………………………………………………………37(*T. geodes*, *T. microsporum*, *T. nubicola*, *T. pseudoalbum*, *T. terricola*, *T. tundrense*, *T. yunnanense*) 34a. Conidia relatively more minor (globose,1.3 µm diam)……………………………………………………….***T. endophyticum*** 34b. Conidia larger, diam > 1.3 µm……………………………………………………………………………….………35 35a. Conidia > 4 µm long (4.5–9.0 × 2.5–3.5 µm)…………..………………………………………………***T. ovalisporum*** 35b. Conidia < 4 µm long……………………………………………………………………………………………….…36 36a. Phialides 4.6 ± 1.2 × 1.5 ± 0.3µm, conidia spherical, larger, 2.1–2.2 µm diam……………………….***T. amazonense*** 36b. Phialide 4.6 × 1.5 µm, conidia spherical, relatively smaller, 1.5 ± 0.1 µm diam………………………………***T. tropicale*** 37a. Phialides cylindrical………………………………………………………………………………………………….38 37b. Phialides subglobose or ellipsoidal…………………………………………………………………………………41 38a. Conidia ellipsoidal, globose or broadly ellipsoidal………………………………………………….……………39 38b. Conidia asymmetrically flattened, with a minute apical……………………………………….…***T. microsporum*** 39a. Colonies white………………………………………………………………………………………………………..40 39b. Colonies white or pale yellow (Phialides 12.3–48.5 × 1.0–2.0 μm, conidia smaller, 1.8–3.4 × 1.3–1.9 μm)………………………………………………………….…………………………………………….***T. pseudoalbum*** 40a. Phialides shorter, 5.6–12.4 × 1.4–2.4 µm, conidia 1.9–2.4 × 1.6–2.0 µm………………………………………………..***T. geodes*** 40b. Phialides longer, 7.6–62.6 × 0.9–2.3 μm, conidia 1.2–2.4 × 0.9–1.9 μm……………………………………..***T. yunnanense*** 41a. Conidia only one type…………………………………………………………………………..……………………42 41b. Conidia two types (microconidia ellipsoidal or reniform, 2.3–4.2 × 1.3–2.3 µm, macroconidia: cylindrical, 10 × 2.4 µm) …………………………………………………………..……………………………………………***T. tundrense*** 42a. Phialides relatively longer, 4.4–7.8 × 1.5–2.7 µm, conidia cylindrical, 2.6–4.1 × 0.8–1.3 µm, colonies white to pale cream……………………………………………………………………………………………………..***T. nubicola*** 42b. Phialides shorter, 2.8–3.5 × 2.0–3.0 µm, conidia broadly oval, 2.5–3 × 2.0–2.5 µm, colonies white….***T. terricola*** 43a. On *Elaphomyces*………………………………………………………………………………………***T. guangdongense*** 43b. From *Ophiocordyceps sinensis*……………………………………………………………………………………………………44 44a. Conidia reniform, 1.0–3.2 × 0.7–1.6 µm, phialides 3.4–10.6 × 1.1–3.8 µm……………………….***T. reniformisporum*** 44b. Conidia spherical, 1.4–3.6 µm diam, phialides 7.6–19.4 × 2.9–3.6 µm.……………………………………***T. sinense*** 45a. On mosquito larvae, conidia two types (ellipsoidal: 2–2.5 × 1.5–2 µm, subglobose to ellipsoidal, or kidney-shaped: 3.5–4 × 3–3.5 µm)…………………………………………………………………………….. ***T. extinguens*** 45b. On bdelloid rotifers, conidia only one type……………………………………………………..…………………46 46a. Phialides thicker, 4–8 × 3–4.5 µm, conidia circular, 2.5–3.2 × 1.5–2.0 µm, colonies pure white………***T. lignicola***
46b. Phialides slender, 4.8–9.8 × 1.4–3.5 µm, conidia like an equilateral triangle or less ellipsoidal, 2–3 × 1.3–1.7 µm, colonies white or pale yellow……………………………………………………………………………………***T. trigonosporum***

## 4. Discussion

*Tolypocladium* is one of the most diverse fungal groups in terms of shape, substrate or host, and habitat range. Many new species have recently been added to *Tolypocladium* [11,12,13,14,73]. The present study described three new species (*T*. *pseudoalbum* sp. nov., *T*. *subparadoxum* sp. nov., and *T*. *yunnanense* sp. nov.) based on phylogenetic analyses and morphological characteristics. Phylogenetically, these three species fell within the *Tolypocladium* clade, while morphologically all three species possessed cylindrical phialides and ellipsoidal or globose conidia. It is challenging to distinguish species of *Tolypocladium* based only on morphological characteristics, because several species in this genus are morphologically cryptic [7,8,11]. Sexual morphological features are diverse: the ovoid perithecia may be superficial or completely immersed and part spores size varies [7,10]. However, the asexual morphological features are relatively simple.

Species of *Tolypocladium* play a significant role in a variety of artificial and wild ecosystems and may participate in antifungal, host–fungi, and insecticidal interactions [10,77]. Many species have been described in *Tolypocladium* based on host associations or morphology [11,12]. Over the past several decades, the increasing number of new fungal species being discovered globally has dramatically changed the classification of early-diverging fungi [78]. In most previous studies, the classification of *Tolypocladium* was developed based on morphological characteristics. However, the advent of molecular biology, which was an important scientific milestone, revolutionized the taxonomic characterization of this genus. Over the last few decades, the number of accepted species in *Tolypocladium* has doubled.

All 48 of the currently accepted species of *Tolypocladium* were included in the key developed in this study. However, because the sequence loci for many of these taxa were incomplete, only 27 species were included in the multigene phylogenetic analyses (Figure 1). The multilocus phylogenetic approach used in this study of the genus *Tolypocladium* shed considerable light on this influential group of fungi.

The ITS region is the most commonly used molecular marker for species delimitation in fungi. Schoch et al. proposed ITS as the standard barcode for fungi. That proposal will satisfy most fungal biologists, but not all [57,79,80]. Species-level identification of fungi has long been considered challenging. Carlson et al. reported that ITS has a low molecular variation in *Trametes* leading to poorly resolved phylogenies and unclear species boundaries, especially in the *T. versicolor* species complex [80]. The results of this study indicated that the ITS sequences did not help substantially to separate *Tolypocladium* species. However, the ITS sequences did help to resolve the phylogenetic relationships between *Tolypocladium* and related genera. The analyses of molecular phylogeny based on ITS sequences used in the current classification of the genus fungus are congruent with the higher genus clades inferred from these analyses. However, ITS sequence data are not likely to resolve species-level relationships or to delimitate closely related species and species complexes. Using the ITS phylogeny, it was still not possible to identify some species of *Tolypocladium* with confidence in the new classification system; the ITS region alone could not accurately identify species in *Tolypocladium.* For example, in the ITS phylogeny, *T*. *varium* CBS 429.94 was inseparable from *T*. *inflatum* OSC 71235 and *T*. *inflatum* NBRC 31669, while *T*. *tundrense* CBS 569.84 was inseparable from *T*. *cylindrosporum* ARSEF 2920 and *T*. *cylindrosporum* YFCC 1805001 (Figure 2). In contrast, relationships among *Tolypocladium* species were highly resolved in the phylogeny based on the protein-coding gene *rpb1.* Multilocus sequence analyses provide additional information to better characterize species boundaries [81]. Therefore, we used both morphological and multilocus phylogenetic evidence to support the novelty of the new species described in this study and to ensure accurate species identifications.

*Tolypocladium extinguens* was first reported from New Zealand by Samson et al. The original description was based on only a single isolate [82]. *Tolypocladium extinguens* is characterized by its prolonged growth in pure culture and its subglobose to ellipsoidal, sometimes kidney-shaped, conidia [82]. Our phylogenetic analysis did not support the placement of this species in *Tolypocladium* due to long branch attraction in the phylogenetic tree. More taxa must be added to this analysis in future to clarify the phylogenetic position of this species.

*Tolypocladium* species are well-known medicinal fungi that are also plant endophytes, soil inhabitants, and insect pathogens [10,12]. Because many of species of fungi are present in the soil environment at some stage of their life cycle, this substrate is preferred by researchers for the isolation of *Tolypocladium.* At least eight species have been reported from the soil: *T. geodes*, *T. microsporum*, *T. nubicola*, *T. pseudoalbum* sp. nov., *T. subparadoxum* sp. nov., *T. terricola*, *T. tundrense*, and *T. yunnanense* sp. nov. In Asia (China, Japan, and Thailand), *Tolypocladium* species are mainly known from insects [19], and few studies have focused on *Tolypocladium* species in the soil and in plant roots. Recently, *Tolypocladium* species in Chinese soils were surveyed, but no new species were identified.

## Figures and Tables

**Figure 1 jof-08-01158-f001:**
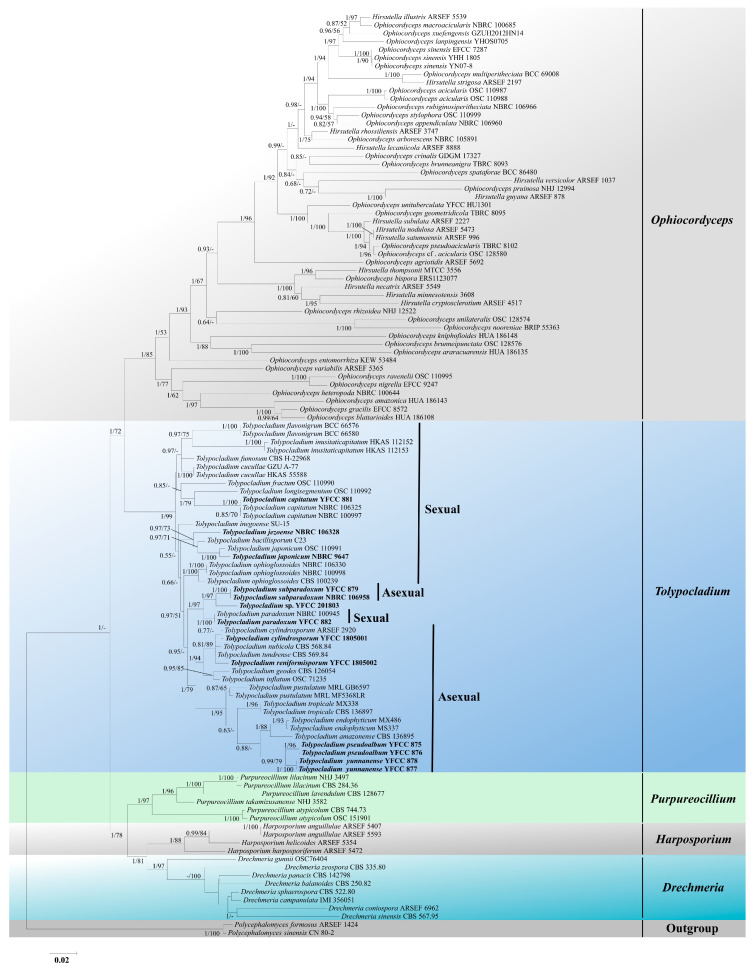
Maximum-likelihood tree illustrating the phylogeny of *Tolypocladium* based on the combined dataset of nr*SSU*, nr*LSU*, *tef*-*1α*, *rpb1* and *rpb2* sequences. *Polycephalomyces formosus* ARSEF 1424 and *Polycephalomyces sinensis* CN 80-2 were used as outgroups. The maximum-likelihood bootstrap values (≥50) and Bayesian posterior probability values (≥0.50) are indicated above the branches. Isolates in bold type are those analyzed in this study.

**Figure 2 jof-08-01158-f002:**
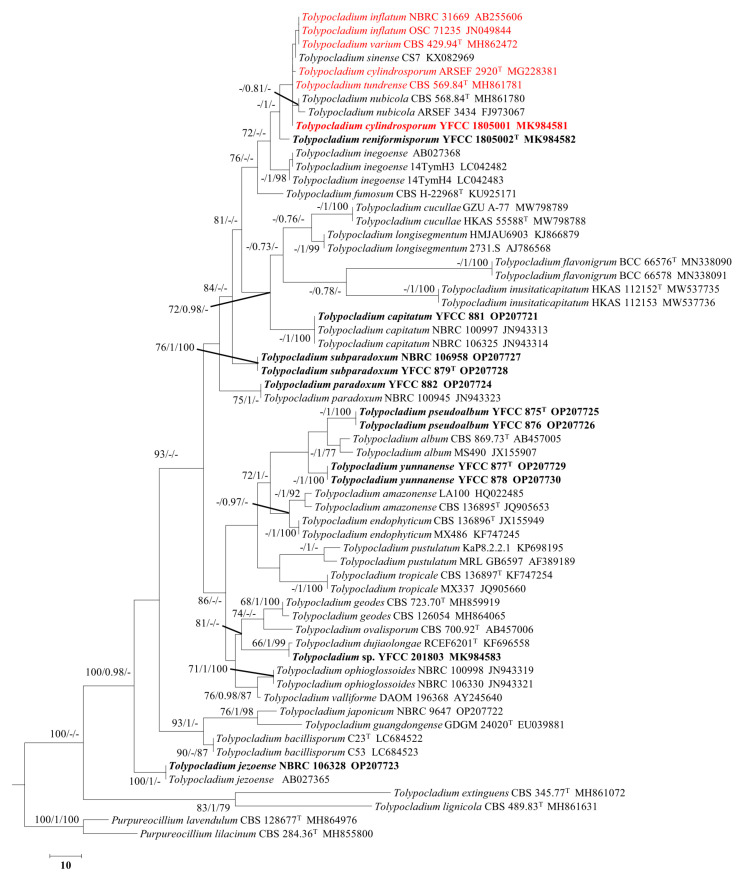
Maximum parsimony, Bayesian analysis, and RAxML tree illustrating the phylogeny of *Tolypocladium* derived from ITS sequences. Statistical support values (MP bootstrap/Bayesian posterior probability/ML bootstrap ≥ 70%) are shown at the nodes. The indistinguishable species are in red and the isolates analyzed in this study are in bold.

**Figure 3 jof-08-01158-f003:**
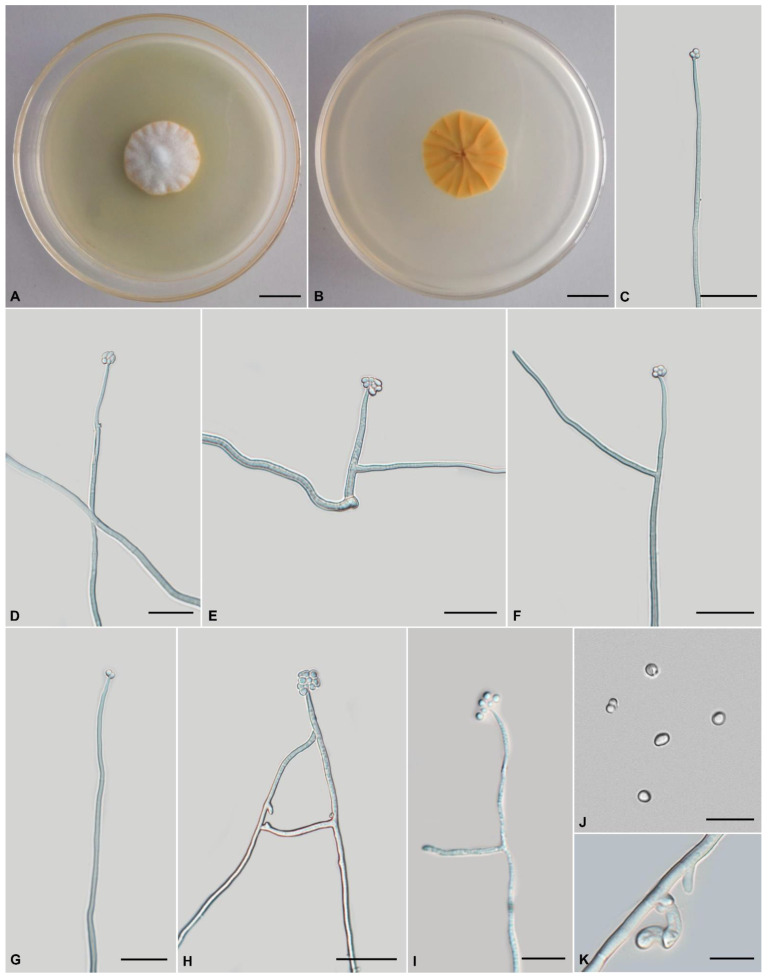
Morphology of *Tolypocladium pseudoalbum* (YFCC 875, ex-type living culture). (**A**,**B**) Culture characteristics on PDA medium incubated at 22 °C for 14 days; (**C**–**I**) phialides; (**J**) conidia; (**K**) chlamydospore. Scale bars: (**A**,**B**) = 10 mm; (**C**–**H**) = 20 μm; (**I**–**K**) = 10 μm.

**Figure 4 jof-08-01158-f004:**
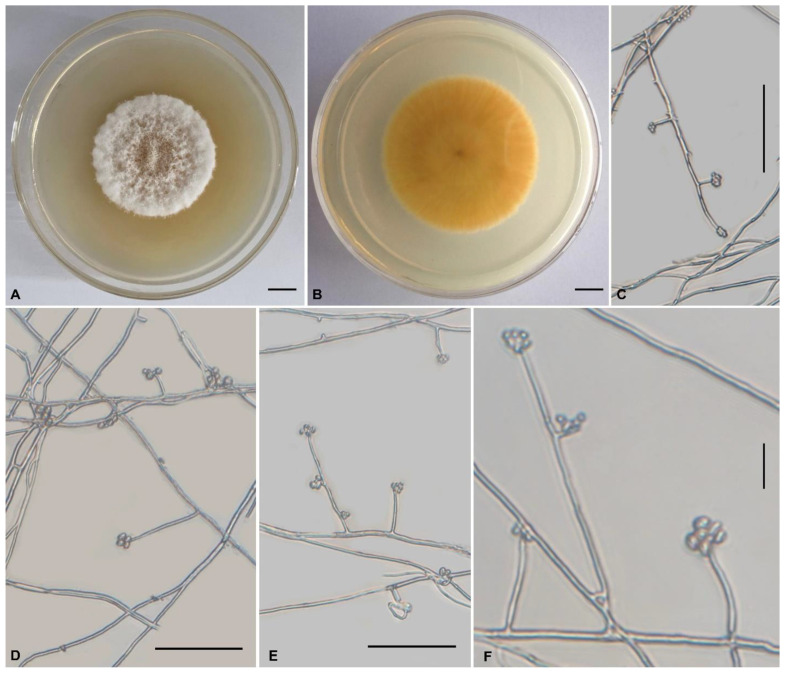
Morphology of *Tolypocladium subparadoxum* (YFCC 879, ex-type living culture). (**A**,**B**) Culture characteristics on PDA medium incubated at 22 °C for 21 days; (**C**–**F**) phialides and conidia. Scale bars: (**A**,**B**) = 10 mm; (**C**–**E**) = 50 μm; (**F**) = 20 μm.

**Figure 5 jof-08-01158-f005:**
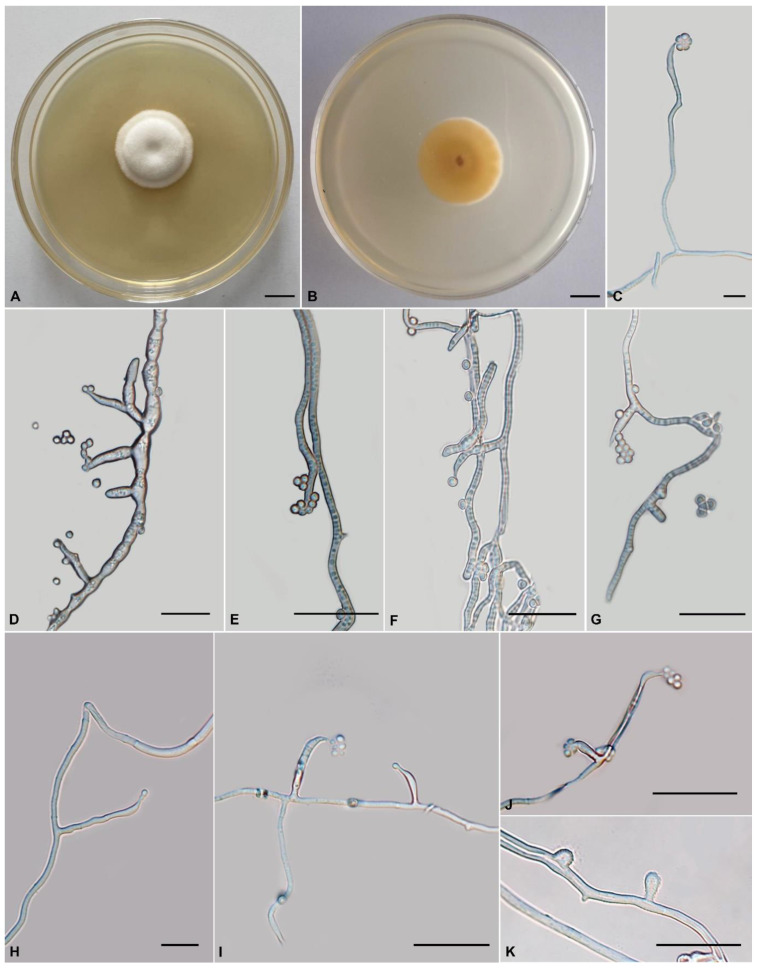
Morphology of *Tolypocladium yunnanense* (YFCC 877, ex-type living culture). (**A**,**B**) Culture characteristics on PDA medium incubated at 22 °C for 14 days; (**C**–**J**) phialides and conidia; (**K**) chlamydospore. Scale bars: A–B = 10 mm; C, H = 10 μm; D–G, I–K = 20 μm.

**Table 1 jof-08-01158-t001:** Specimen information and GenBank accession numbers of sequences used in this study.

Taxon	Voucher Information	GenBank Accession Number					Reference
		nr*SSU*	nr*LSU*	*tef-1α*	*rpb1*	*rpb2*	
*Drechmeria balanoides*	CBS 250.82^T^	AF339588	AF339539	DQ522342	DQ522388	DQ522442	[47,48]
*Drechmeria campanulata*	IMI 356051^T^	AF339592	AF339543	-	-	-	[47]
*Drechmeria coniospora*	ARSEF 6962	-	LAYC01000003	LAYC01000001	LAYC01000003	LAYC01000002	[49]
*Drechmeria gunnii*	OSC 76404	AF339572	AF339522	AY489616	AY489650	DQ522426	[7,47]
*Drechmeria panacis*	CBS 142798^T^	MF588890	MF588897	MF614144	-	-	[50]
*Drechmeria sinensis*	CBS 567.95	AF339594	AF339545	DQ522343	DQ522389	DQ522443	[47,48]
*Drechmeria sphaerospora*	CBS 522.80^T^	AF339590	AF339541	-	-	-	[47]
*Drechmeria zeospora*	CBS 335.80^T^	AF339589	AF339540	EF469062	EF469091	EF469109	[7,47]
*Harposporium anguillulae*	ARSEF 5407	-	AY636080	-	-	-	[51]
*Harposporium anguillulae*	ARSEF 5593	-	AY636081	-	-	-	[51]
*Harposporium harposporiferum*	ARSEF 5472^T^	AF339569	AF339519	DQ118747	DQ127238	-	[47,48]
*Harposporium helicoides*	ARSEF 5354	AF339577	AF339527	-	-	-	[47]
*Hirsutella citriformis*	ARSEF 1446	KM652065	KM652106	KM651990	KM652031	-	[52]
*Hirsutella cryptosclerotium*	ARSEF 4517^T^	KM652066	KM652109	KM651992	KM652032	-	[52]
*Hirsutella fusiformis*	ARSEF 5474	KM652067	KM652110	KM651993	KM652033	-	[52]
*Hirsutella guyana*	ARSEF 878	KM652068	KM652111	KM651994	KM652035	-	[52]
*Hirsutella illustris*	ARSEF 5539	KM652069	KM652112	KM651996	KM652037	-	[52]
*Hirsutella lecaniicola*	ARSEF 8888	KM652071	KM652114	KM651998	KM652038	-	[52]
*Hirsutella minnesotensis*	3608	JPUM01000376	JPUM01000376	JPUM01000211	JPUM01000139	JPUM01000138	[53]
*Hirsutella necatrix*	ARSEF 5549	KM652073	KM652116	KM651999	KM652039	-	[52]
*Hirsutella nodulosa*	ARSEF 5473	KM652074	KM652117	KM652000	KM652040	-	[52]
*Hirsutella radiate*	ARSEF 1369	KM652076	KM652119	KM652002	KM652042	-	[52]
*Hirsutella rhossiliensis*	ARSEF 3747	KM652080	KM652123	KM652006	KM652045	-	[52]
*Hirsutella satumaensis*	ARSEF 996	KM652082	KM652125	KM652008	KM652047	-	[52]
*Hirsutella strigose*	ARSEF 2197	KM652085	KM652129	KM652012	KM652050	-	[52]
*Hirsutella subulata*	ARSEF 2227	KM652086	KM652130	KM652013	KM652051	-	[52]
*Hirsutella thompsonii*	MTCC 3556	APKB01000383	APKB01000383	APKB01000061	APKB01000125	APKB01000164	[54]
*Hirsutella versicolor*	ARSEF 1037	KM652102	KM652150	KM652029	KM652063	-	[52]
*Ophiocordyceps acicularis*	OSC 110987	EF468950	EF468805	EF468744	EF468852	-	[7]
*Ophiocordyceps acicularis*	OSC 110988	EF468951	EF468804	EF468745	EF468853	-	[7]
*Ophiocordyceps agriotidis*	ARSEF 5692	DQ522540	DQ518754	DQ522322	DQ522368	DQ522418	[48]
*Ophiocordyceps amazonica*	HUA 186143^T^	KJ917562	KJ917571	KM411989	KP212902	KM411982	[55]
*Ophiocordyceps appendiculata*	NBRC 106960	JN941728	JN941413	AB968577	JN992462	AB968539	[56,57]
*Ophiocordyceps arborescens*	NBRC 105891^T^	AB968386	AB968414	AB968572	-	AB968534	[56]
*Ophiocordyceps bispora*	ERS1123077	FKNF01000183	FKNF01000183	FKNF01000002	FKNF01000038	FKNF01000031	[58]
*Ophiocordyceps blattarioides*	HUA 186108^T^	KJ917558	KJ917569	-	KP212912	KM411984	[55]
*Ophiocordyceps brunneanigra*	TBRC 8093^T^	-	MF614654	MF614638	MF614668	MF614681	[59]
*Ophiocordyceps brunneipunctata*	OSC 128576	DQ522542	DQ518756	DQ522324	DQ522369	DQ522420	[48]
*Ophiocordyceps* cf. *acicularis*	OSC 128580	DQ522543	DQ518757	DQ522326	DQ522371	DQ522423	[48]
*Ophiocordyceps crinalis*	GDGM 17327	KF226253	KF226254	KF226256	KF226255	-	[60]
*Ophiocordyceps entomorrhiza*	KEW 53484	EF468954	EF468809	EF468749	EF468857	EF468911	[7]
*Ophiocordyceps geometridicola*	TBRC 8095^T^	-	MF614648	MF614632	MF614663	MF614679	[59]
*Ophiocordyceps gracilis*	EFCC 8572	EF468956	EF468811	EF468751	EF468859	EF468912	[7]
*Ophiocordyceps heteropoda*	NBRC 100644	JN941718	JN941423	AB968596	JN992452	AB968557	[56,57]
*Ophiocordyceps kniphofioides*	HUA 186148	KC610790	KF658679	KC610739	KF658667	KC610717	[55]
*Ophiocordyceps lanpingensis*	YHOS0705	KC417458	KC417460	KC417462	KC417464	KC456333	[61]
*Ophiocordyceps macroacicularis*	NBRC 100685^T^	AB968388	AB968416	AB968574	-	AB968536	[56]
*Ophiocordyceps multiperitheciata*	BCC 69008^T^	-	MF614657	MF614641	-	MF614682	[59]
*Ophiocordyceps nigrella*	EFCC 9247	EF468963	EF468818	EF468758	EF468866	EF468920	[7]
*Ophiocordyceps nooreniae*	BRIP 55363^T^	KX673811	KX673810	KX673812	-	KX673809	[62]
*Ophiocordyceps pseudoacicularis*	TBRC 8102^T^	-	MF614646	MF614630	MF614661	MF614677	[59]
*Ophiocordyceps pruinosa*	NHJ 12994	EU369106	EU369041	EU369024	EU369063	EU369084	[63]
*Ophiocordyceps ravenelii*	OSC 110995	DQ522550	DQ518764	DQ522334	DQ522379	DQ522430	[48]
*Ophiocordyceps rhizoidea*	NHJ 12522	EF468970	EF468825	EF468764	EF468873	EF468923	[7]
*Ophiocordyceps rubiginosiperitheciata*	NBRC 106966	JN941704	JN941437	AB968582	JN992438	AB968544	[56,57]
*Ophiocordyceps sinensis*	EFCC 7287	EF468971	EF468827	EF468767	EF468874	EF468924	[7]
*Ophiocordyceps sinensis*	YN07-8	JX968027	JX968032	JX968017	JX968007	JX968012	[64]
*Ophiocordyceps sinensis*	YHH 1805	MK984568	MK984580	MK984572	MK984587	MK984576	[11]
*Ophiocordyceps spataforae*	BCC 86480^T^	-	MG831747	MG831746	MG831748	MG831749	[59]
*Ophiocordyceps stylophora*	OSC 110999	EF468982	EF468837	EF468777	EF468882	EF468931	[7]
*Ophiocordyceps unilateralis*	OSC 128574	DQ522554	DQ518768	DQ522339	DQ522385	DQ522436	[48]
*Ophiocordyceps unituberculata*	YFCC HU1301^T^	KY923214	KY923212	KY923216	KY923218	KY923220	[65]
*Ophiocordyceps variabilis*	ARSEF 5365	DQ522555	DQ518769	DQ522340	DQ522386	DQ522437	[48]
*Ophiocordyceps xuefengensis*	GZUH2012HN14^T^	KC631789	-	KC631793	KC631798	-	[66]
*Polycephalomyces formosus*	ARSEF 1424	KF049615	AY259544	DQ118754	DQ127245	KF049671	[43,51,67,68]
*Polycephalomyces sinensis*	CN 80-2	HQ832887	HQ832886	HQ832890	HQ832888	HQ832889	[69]
*Purpureocillium atypicolum*	CBS 744.73	EF468987	EF468841	EF468786	EF468892	-	[7]
*Purpureocillium atypicolum*	OSC 151901	KJ878914	KJ878880	KJ878961	KJ878994	-	[8]
*Purpureocillium lavendulum*	CBS 128677^T^	-	FR775489	FR775516	FR775512	FR775538	[70]
*Purpureocillium lilacinum*	CBS 284.36^T^	AY526475	FR775484	EF468792	EF468898	EF468941	[7,70,71]
*Purpureocillium lilacinum*	NHJ 3497	EU369096	EU369033	EU369014	EU369053	EU369074	[63]
*Purpureocillium takamizusanense*	NHJ 3582	EU369097	EU369034	EU369015	-	-	[63]
*Tolypocladium amazonense*	CBS 136895^T^	KF747314	KF747134	KF747099	KF747214	-	[72]
*Tolypocladium bacillisporum*	C23	LC684522	LC684522	LC684525			[13]
*Tolypocladium capitatum*	NBRC 100997	JN941740	JN941401	AB968597	JN992474	AB968558	[56,57]
*Tolypocladium capitatum*	NBRC 106325	JN941739	JN941402	AB968598	JN992473	AB968559	[56,57]
*Tolypocladium capitatum*	**YFCC 881**	**OP207711**	**OP207731**	**OP223145**	**OP223123**	**OP223133**	**Present study**
*Tolypocladium cucullae*	GZU A-77	MW798785	MW798787	-	-	-	[73]
*Tolypocladium cucullae*	HKAS 55588	MW798784	MW798786	-	-	-	[73]
*Tolypocladium cylindrosporum*	ARSEF 2920^T^	-	MH871712	MG228390	MG228384	MG228387	[15,74]
*Tolypocladium cylindrosporum*	YFCC 1805001	MK984565	MK984577	MK984569	MK984584	MK984573	[11]
*Tolypocladium endophyticum*	MS337	KF747315	KF747136	KF747101	KF747215	-	[72]
*Tolypocladium endophyticum*	MX486	KF747321	KF747152	KF747116	KF747232	-	[72]
*Tolypocladium flavonigrum*	BCC 66576	-	MN337287	MN338495	-	-	[14]
*Tolypocladium flavonigrum*	BCC 66580	-	MN337289	MN338497	MN338494	-	[14]
*Tolypocladium fractum*	OSC 110990	DQ522545	DQ518759	DQ522328	DQ522373	DQ522425	[48]
*Tolypocladium fumosum*	CBS H-22968^T^	-	KU985053	-	-	-	[75]
*Tolypocladium geodes*	CBS 126054	-	MH875520	-	-	-	[74]
*Tolypocladium inegoense*	SU-15	-	DQ118741	DQ118752	DQ127243	-	[51]
*Tolypocladium inflatum*	OSC 71235	EF469124	EF469077	EF469061	EF469090	EF469108	[7]
*Tolypocladium inusitaticapitatum*	HKAS 112152	MW537733	MW537718	MW507527	-	MW507529	[12]
*Tolypocladium inusitaticapitatum*	HKAS 112153	MW537734	MW537719	MW507528	-	MW507530	[12]
*Tolypocladium japonicum*	**NBRC 9647**	**OP207712**	**OP207732**	**OP223146**	**OP223124**	**OP223134**	**Present study**
*Tolypocladium jezoense*	**NBRC 106328**	**OP207713**	**OP207733**	**OP223147**	**OP223125**	**OP223135**	**Present study**
*Tolypocladium longisegmentum*	OSC 110992	-	EF468816	-	EF468864	EF468919	[7]
*Tolypocladium nubicola*	CBS 568.84^T^	-	MH873478	-	-	-	[74]
*Tolypocladium ophioglossoides*	CBS 100239	KJ878910	KJ878874	KJ878958	KJ878990	KJ878944	[8]
*Tolypocladium ophioglossoides*	NBRC 100998	JN941735	JN941406	AB968602	JN992469	AB968563	[56,57]
*Tolypocladium ophioglossoides*	NBRC 106330	JN941734	JN941407	AB968603	JN992468	AB968564	[56,57]
*Tolypocladium paradoxum*	NBRC 100945	JN941731	JN941410	AB968599	JN992465	AB968560	[56,57]
*Tolypocladium paradoxum*	**YFCC 882**	**OP207714**	**OP207734**	**OP223148**	**OP223126**	**OP223136**	**Present study**
*Tolypocladium pseudoalbum*	**YFCC 875^T^**	**OP207717**	**OP207737**	**OP223151**	**OP223129**	**OP223139**	**Present study**
*Tolypocladium pseudoalbum*	**YFCC 876**	**OP207718**	**OP207738**	**OP223152**	**OP223130**	**OP223140**	**Present study**
*Tolypocladium pustulatum*	MRL GB6597	-	AF389190	-	-	-	[18]
*Tolypocladium pustulatum*	MRL MF5368LR	-	AF373282	-	-	-	[18]
*Tolypocladium reniformisporum*	YFCC 1805002^T^	MK984566	MK984578	MK984570	MK984585	MK984574	[11]
*Tolypocladium* sp.	YFCC 201803	MK984567	MK984579	MK984571	MK984586	MK984575	[11]
*Tolypocladium subparadoxum*	**NBRC 106958**	**OP207715**	**OP207735**	**OP223149**	**OP223127**	**OP223137**	**Present study**
*Tolypocladium subparadoxum*	**YFCC 879^T^**	**OP207716**	**OP207736**	**OP223150**	**OP223128**	**OP223138**	**Present study**
*Tolypocladium tropicale*	CBS 136897^T^	-	KF747125	KF747090	KF747204	-	[72]
*Tolypocladium tropicale*	MX338	KF747318	KF747149	KF747113	KF747229	-	[72]
*Tolypocladium tundrense*	CBS 569.84^T^	-	MH873479	-	-	-	[74]
*Tolypocladium yunnanense*	**YFCC 877^T^**	**OP207719**	**OP207739**	**OP223153**	**OP223131**	**-**	**Present study**
*Tolypocladium yunnanense*	**YFCC 878**	**OP207720**	**OP207740**	**OP223154**	**OP223132**	**-**	**Present study**

Boldface: data generated in this study. ^T^ ex-type material.

## Data Availability

The datasets presented in this study can be found in GenBank. The accession numbers can be found in the article (Table 1).

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
