# Peer review of "Phylogeny and Systematics of the Genus *Tolypocladium* (Ophiocordycipitaceae, Hypocreales)"

_jof, 2022, doi:10.3390/jof8111158_

Round 1

Reviewer 1 Report

The study by Dong et al., conducted the comprehensive phylogenetic analyses and description of three novel species. While the paper contains valuable study results on taxonomy of Tolypocladium, the phylogenetic analysis should be more comprehensive. Furthermore there seems to be some incorrect and insufficient explanations in the taxonomic section. Please find attached pdf file (jof-1917331-peer-review-v1_REV220913.pdf) and revise based on comments.

Author Response

Reply to Reviewer 1:

Point 1: The study by Dong et al., conducted the comprehensive phylogenetic analyses and description of three novel species. While the paper contains valuable study results on taxonomy of Tolypocladium, the phylogenetic analysis should be more comprehensive.

Response 1: Thanks so much for the comprehensive review. More discussion on phylogenetic analysis has been included in the revised manuscript. Also, literature reference has complemented here: (marked highlight yellow, Line 41-43: Cordyceps sensu lato was recently reclassified into three families (Clavicipitaceae sensu stricto, Cordycipitaceae, and Ophiocordycipitaceae) and four genera (Cordyceps s. str., Elaphocordyceps, Metacordyceps, and Ophiocordyceps). (lines 40-42, lines 206-214)

Point 2: Furthermore there seems to be some incorrect and insufficient explanations in the taxonomic section. Please find attached pdf file (jof-1917331-peer-review-v1_REV220913.pdf) and revise based on comments.

Response 2: It’s very kind of the reviewer to help us point out this mistake, we have corrected the erroneous. (lines 283-284)

Reviewer 2 Report

Dear Editor, I have reviewed the manuscript entitled 

Phylogeny and systematics of the genus Tolypocladium (Ophiocordycipitaceae, Hypocreales), by authors Quan-Ying, Dong et al., which describes a study of the genus Tolypocladium, which is part of the order Hypocreales.

The manuscript is well written and I find it very well presented. 

It is a global work, which includes a phylogenetic analysis and also presents a detailed dichotomous key for taxonomic identification of species based on morphological attributes.

This is an important contribution, since in general, phylogenetic studies do not provide the keys that allow understanding the differences between species that are separated with molecular tools.

I also find the information on materials and method adequate, and it is clear that the authors know this order of fungi well.

As part of the results, three new species are described, which were also cultivated and described in their morphological attributes, these are Tolypocladium pseudoalbum sp. nov., Tolypocladium subparadoxum sp. nov., and Tolypocladium yunnanense sp. nov., .

I find the discussion interesting, however I believe that there are other publications that have also shown that ITS is not sufficient to separate fungal species, so I suggest looking for more information on this subject (line 474-489), to complement or to give more strength to the statements  that are made.

I suggest that the article be accepted and I do not find any editorial problems that would warrant any corrections.

Sincerely, 

Author Response

Reply to Reviewer 2:

Dear Editor, I have reviewed the manuscript entitled

Phylogeny and systematics of the genus Tolypocladium (Ophiocordycipitaceae, Hypocreales), by authors Quan-Ying, Dong et al., which describes a study of the genus Tolypocladium, which is part of the order Hypocreales.

The manuscript is well written and I find it very well presented.

It is a global work, which includes a phylogenetic analysis and also presents a detailed dichotomous key for taxonomic identification of species based on morphological attributes.

This is an important contribution, since in general, phylogenetic studies do not provide the keys that allow understanding the differences between species that are separated with molecular tools.

I also find the information on materials and method adequate, and it is clear that the authors know this order of fungi well.

As part of the results, three new species are described, which were also cultivated and described in their morphological attributes, these are Tolypocladium pseudoalbum sp. nov., Tolypocladium subparadoxum sp. nov., and Tolypocladium yunnanense sp. nov., .

I find the discussion interesting, however I believe that there are other publications that have also shown that ITS is not sufficient to separate fungal species, so I suggest looking for more information on this subject (line 474-489), to complement or to give more strength to the statements that are made.

I suggest that the article be accepted and I do not find any editorial problems that would warrant any corrections.

Sincerely,

Response :

Thank you so much, I cannot imagine how the audience in the community might be misled should the manuscript be published without adoption of this pertinent advice, in the revised manuscript, more related information on this subject have been supplemented in addition to the phylogenetic analysis based on ITS. (lines 496-501 and lines 515-516)

Round 2

Reviewer 1 Report

Figs 1-5

The authors missed my comment in the former review report. Please respond to the following comments:

Fig. 1:

Although some species with only one or two gene regions are included in the dataset (e.g. T. cucullae, T. longisegmentum, and T. fumosum), some are not included without indicating the reason. In addition, T. bacillisporum is not included despite nrSSU, ITS, nrLSU and tef-1α sequence present in Genbank.

Fig. 2:

Please add T. bacillisporum for comprehensive phylogeny.

Figs 3-5

The authors missed my comment in the former review report. Please add culture ID in the caption for these figure.

Page 12, lines 300-307 & Fig. 2

The authors identified “NBRC 106958, Niryo, Takatsuki-shi, Osaka Pref” as Tolypocladium subparadoxum and mentioned that T. subparadoxum only observed as anamorph. This interpretation is clearly incorrect: NBRC 106958 is isolated from teleomorph (ascospore) and already identified as Tolypocladium paradoxum by S. Ban (please see: https://www.nite.go.jp/nbrc/catalogue/NBRCCatalogueDetailServlet?ID=NBRC&CAT=00106958). Because morphology of anamorph of T. paradoxum is unknown at this point, there is no way to compare T. subparadoxum and T. paradoxum in morphology. In other words, the possibility that T. subparadoxum is a synonym of T. paradoxum cannot be ruled out.

I checked ITS homology between T. paradoxum NBRC 100945 and T. paradoxum NBRC 106958: they show low homology (95.1%, 525/552). Thus NBRC 100945 and 106958 is probably different species and at least either one is not T. paradoxum. If you regard YFCC 879 and NBRC 106958 as a new species T. subparadoxum, it is necessary to explain why NBRC 106958 is not T. paradoxum.

Author Response

Reply to Reviewer 1:

Figs 1-5

The authors missed my comment in the former review report. Please respond to the following comments:

Point 1: Fig. 1: Although some species with only one or two gene regions are included in the dataset (e.g. T. cucullae, T. longisegmentum, and T. fumosum), some are not included without indicating the reason. In addition, T. bacillisporum is not included despite nrSSU, ITS, nrLSU and tef-1α sequence present in Genbank.

Response 1:

Thank you very much for all invaluable comments. It’s our fault that we didn’t notice the comments (7 pieces) inserted along with the highlights in yellow background, which is why the revised manuscript last time didn’t reflect the due changes in response to those comments.

T. bacillisporum has now been included in Fig. 1 for multigene phylogeny analysis (nrSSU, nrLSU, tef-, rpb1, and rpb2). Reasons for species not included in the dataset are justified in section 4. Discussion (lines 507-511).

Point 2: Fig. 2: Please add T. bacillisporum for comprehensive phylogeny.

Response 2:

For T. bacillisporum, in addition to the multigene phylogeny analysis of nrSSU, nrLSU, tef-, rpb1, and rpb2 as illustrated in Fig. 1, the single-gene phylogeny of ITS is updated in Fig. 2.

Point 3: Figs 3-5 The authors missed my comment in the former review report. Please add culture ID in the caption for these figures

Response 3:

Culture ID has now been added in the captions for the following figures:

Figure 3. Morphology of Tolypocladium pseudoalbum (YFCC 875, ex-type living culture) (line 278)

Figure 4. Morphology of Tolypocladium subparadoxum (YFCC 879, ex-type living culture) (line 344)

Figure 5. Morphology of Tolypocladium yunnanense (YFCC 877, ex-type living culture) (line 360)

Point 4: Page 12, lines 300-307 & Fig. 2

The authors identified “NBRC 106958, Niryo, Takatsuki-shi, Osaka Pref” as Tolypocladium subparadoxum and mentioned that T. subparadoxum only observed as anamorph. This interpretation is clearly incorrect: NBRC 106958 is isolated from teleomorph (ascospore) and already identified as Tolypocladium paradoxum by S. Ban (please see: https://www.nite.go.jp/nbrc/catalogue/NBRCCatalogueDetailServlet?ID= NBRC&CAT=00106958). Because morphology of anamorph of T. paradoxum is unknown at this point, there is no way to compare T. subparadoxum and T. paradoxum in morphology. In other words, the possibility that T. subparadoxum is a synonym of T. paradoxum cannot be ruled out.

I checked ITS homology between T. paradoxum NBRC 100945 and T. paradoxum NBRC 106958: they show low homology (95.1%, 525/552). Thus NBRC 100945 and 106958 is probably different species and at least either one is not T. paradoxum. If you regard YFCC 879 and NBRC 106958 as a new species T. subparadoxum, it is necessary to explain why NBRC 106958 is not T. paradoxum.

Response 4:

Thank you very much for the heuristic observation. Given your information, we have modified the teleomorph description of Tolypocladium subparadoxum from “Unknown” to “Not observed” (line 288). The context was that, the culture YFCC 879 was isolated from a soil sample in Simao district, Pu’er city, which was then identified as the same species with NBRC 106958 by multigene phylogeny analyses. For that reason, the strain NBRC 106958 was procured, whereas the specimen NBRC H-12618 was unavailable, and that’s exactly why the teleomorph was “not observed” in this study. Nevertheless, it will no doubt be a part of the subject of our future studies should we observe the teleomorph on T. subparadoxum.

Yes, according to the multigene phylogenic trees presented in this study, YFCC 879 and NBRC 106958 share the same unique position, which are both distinct from another shared position as indicated by YFCC 882 and NBRC 100945, at this point, either position must be qualified to be new species. Concerning Tolypocladium paradoxum NBRC 106958, historically, NBRC 106958 was preceded by NBRC 100945 in isolated year, at this point, NBRC 100945 and YFCC 882 could be more pertinent to be regarded as T. paradoxum. In addition, morphologically, NBRC 106958 distinguishing from NBRC 100945 in size of phialides and conidia, comparisons between T. subparadoxum and T. paradoxum are given in the commentary on T. subparadoxum (see lines 301-323). NBRC 106958 and YFCC 879 were hereby refined as a new species, which was named Tolypocladium subparadoxum.

Round 3

Reviewer 1 Report

The authors revised the manuscript based on the reviewers' comments. However, a small number of additional data are required for acceptance. The authors mention morphological difference between YFCC 879 and NBRC 106958 (line 307), and also between T. subparadoxum and T. paradoxum (lines 311-314)  but it would be easier for the reader to understand if there were photographs of NBRC 106958 and anamorph of T. paradoxum. Especially, since there are no previous studies that have clarified the morphology of T. paradoxum anamorph, it is useful to show its morphology. These photograph may be shown in the supplemental materials rather than in the text.

Author Response

Reply to Reviewer 1:

Point : The authors revised the manuscript based on the reviewers' comments. However, a small number of additional data are required for acceptance. The authors mention morphological difference between YFCC 879 and NBRC 106958 (line 307), and also between T. subparadoxum and T. paradoxum (lines 311-314) but it would be easier for the reader to understand if there were photographs of NBRC 106958 and anamorph of T. paradoxum. Especially, since there are no previous studies that have clarified the morphology of T. paradoxum anamorph, it is useful to show its morphology. These photograph may be shown in the supplemental materials rather than in the text.

Response: Thank you for the patience and this considerate advice, yes, additional photographs of T. subparadoxum NBRC 106958 and anamorph of T. paradoxum NBRC 100945 would be very helpful to clarify the respective morphology, and rationality of the entire study could thus be perceived much more concrete to the readers. It's another considerate piece of advice to add such photographs in supplemental materials rather than the text, see Supplementary Figure 1. as enclosed, we have also noted "(Supplementary Figure 1)" in Line 307 and Line 314 of the last manuscript.

Again, thank you and best regards.